# Validation of the Romanian Version of the Toronto Empathy Questionnaire (TEQ) among Undergraduate Medical Students

**DOI:** 10.3390/ijerph182412871

**Published:** 2021-12-07

**Authors:** Sorin Ursoniu, Costela Lacrimioara Serban, Catalina Giurgi-Oncu, Ioana Alexandra Rivis, Adina Bucur, Ana-Cristina Bredicean, Ion Papava

**Affiliations:** 1Department of Functional Sciences, Discipline of Public Health, Center for Translational Research and Systems Medicine, “Victor Babeș” University of Medicine and Pharmacy, 300041 Timișoara, Romania; sursoniu@umft.ro (S.U.); costela.serban@umft.ro (C.L.S.); 2Department of Neuroscience, Discipline of Psychiatry, Center for Cognitive Research in Neuropsychiatric Pathology (NeuroPsy-Cog), “Victor Babeș” University of Medicine and Pharmacy, 300041 Timișoara, Romania; catalina.giurgi@umft.ro (C.G.-O.); bredicean.ana@umft.ro (A.-C.B.); papava.ion@umft.ro (I.P.); 3Department of Neuroscience, “Carol Davila” University of Medicine and Pharmacy, 020021 Bucharest, Romania; ioana.rivis@gmail.com; 4Psychiatry Compartment, “Dr. Victor Popescu” Emergency Military Clinical Hospital, 300080 Timișoara, Romania

**Keywords:** empathy, Toronto empathy questionnaire (TEQ), medical students, validation

## Abstract

Medical professionals require adequate abilities to identify others’ emotions and express personal emotions. We aimed to determine the validity and reliability of an empathy measuring tool in medical students for this study. We employed Spreng’s Toronto Empathy Questionnaire (TEQ) as a starting point for this validation. The process was performed in several steps, including an English-Romanian-English translation and a focus group meeting to establish each question’s degree of understandability and usability, with minor improvements of wording in each step. We checked internal and external consistency in a pilot group (*n* = 67). For construct and convergent validity, we used a sample of 649 students. The overall internal and external reliability performed well, with Cronbach’s alpha = 0.727 and respective ICC = 0.776. The principal component analysis resulted in 3 components: prosocial helping behavior, inappropriate sensitivity, dismissive attitude. Component 1 includes positively worded questions, and components 2 and 3 include negatively worded questions. Women had significantly higher scores than men in convergent validity, but we did not highlight any differences for other demographic factors. The Romanian version of the TEQ is a reliable and valid tool to measure empathy among undergraduate medical students that may be further used in subsequent research.

## 1. Introduction

Communication forms the basis of interconnection, indispensable for the survival of humankind, allowing for a satisfying physical, mental and social life [1], albeit through verbal, visual, tactile, or, more recently, artificial methods. In our current multifaceted and interconnected society, dominated by many different types of struggles, effective, adequate, and easy communication is essential. To this aim, we must think about what makes for an effective style of communication to benefit everyday interrelationships, especially when it comes to healthcare and the training of future health professionals. As a multidimensional and essential trait of adequate human connections, empathy has been at the forefront of psychological research since the end of the nineteenth century [2]. Concerning its etymology, the term empathy represents the English translation (Edward Titchener, 1909) of the concept of “Einfühlung’’ that was first coined in German by Robert Vischerin 1873 [3], meaning a “feeling into” referring to the psychological mechanism of perspective-taking, namely of projecting oneself into another body or environment and the associated feelings [4]. The skill of perspective-taking, of correctly identifying what other beings are feeling, understanding, and expressing those sentiments, is considered essential for adequate interpersonal communication and maintaining relationships.

Excessive use of social networking causes a decrease in the ability to develop and maintain adequate human interactions, as well as a reduction in the ability to identify the emotions and thoughts of another person (Theory of Mind), and a decrease in the expression of our own emotions (alexithymia) [5]. Medical professionals require the ability to identify emotions (ToM) and express personal emotions (alexithymia); having these skills may determine appropriate empathy levels towards a suffering person. Consequently, constant social media users may show lower levels of empathy as opposed to those who have not constantly been exposed to different imagery and other people’s actions [6].

Empathy represents the ability to attribute mental states to other beings and respond appropriately to them [7], deeming it essential for the successful development and maintenance of social bonds.

Empathy is the basis for positive human interaction and consists of two aspects, namely the affective and cognitive components. The affective component helps us in our emotional responses to others, while the cognitive component entails a rational understanding of another person’s emotional state [8].

There are three types of empathy: cognitive, emotional, and “nurturant” or compassionate empathy [9,10]. Cognitive empathy helps us understand what others are going through, without making their burdens our own [11]. Emotional empathy makes us feel what other people are feeling, while compassionate empathy makes us take action and help the other person deal with their problem. Cognitive empathy is essential in developing the other two, while emotional empathy is not needed in developing compassionate empathy [12]. When dealing with patients, cognitive empathy is valued, therefore the term “clinical empathy” has been established to better describe that doctor-patient relationship [13]. Clinical empathy comprises and goes beyond understanding the patients point of view and feelings, to communicating and making sure that the perceived feelings/situations are accurate and finally, taking action to help the patient [14]. “Nurturant” empathy is characterized by the fact that the medical professional is supportive to the patient, providing total security and attention [15].

Empathy determines an altruistic behavior to help others when suffering, which is an essential human trait that we aim to develop in medical students. The medical profession is often described as burdensome, given that there are long working hours, high pressure, and stress. Therefore, burnout represents a real threat amongst medical professionals, often determined by compassion fatigue, which is in close relation to empathy [16]. The patient-doctor relationship is reliant on empathy, as it helps build trust, enforces better communication between the two, cultivating a safe environment in order to evaluate possibilities and make the best medical decisions [17,18]. Having a doctor with high levels of empathy has been proven to determine a decrease in stress levels in patients, diminish anxiety, depression and improve the prognosis [19,20].

Empathy contributes to establishing a relationship based on trust between the healthcare professional and the patient, enhancing treatment adherence, diagnostic accuracy, and patient satisfaction [21,22]. Having an empathic doctor does not only benefit the patient but the healthcare worker himself, as well. Treatment adherence and patient satisfaction increase job satisfaction and reduce the feeling of burnout [23,24].

The Toronto empathy questionnaire (TEQ), developed by Spreng et al. [1], is a unidimensional, brief, and valid instrument for the assessment of empathy. TEQ comprises 16 items represented by questions. Answers must be chosen from the following options: Never = 0; Rarely = 1; Sometimes = 2; Often = 3; Always = 4. Item responses are scored according to the following scale for positively worded items: 1, 3, 5, 6, 8, 9, 13, 16. The following negatively worded items are reverse scored: 2, 4, 7, 10, 11, 12, 14, 15. Total scores represent the sum of all items [1].

Considering the highly significant role played by empathy in the context of a sanogenic therapeutic alliance, benefitting the service-users and health professionals alike [25] we sought to examine this ability in a sample of undergraduate medical students, with a view of contributing to the existing international body of literature dedicated to this area of research interest, as well as suggesting helpful development strategies that could be included in the medical curricula.

More, specifically, this study aimed to determine the validity and reliability of TEQ among undergraduate medical students in Romania to be further used in subsequent research.

This work is part of a larger project, with a purpose to identify the connection between the use of social networking and individual Theory of Mind (ToM), alexithymia, and empathy levels of students of the Faculty of General Medicine of the “Victor Babes” University of Medicine and Pharmacy Timisoara, Romania, to raise awareness about this new type of addiction.

## 2. Materials and Methods

### 2.1. Preliminary Preparations

This study was approved by the Research Ethics Committee of the “Victor Babes” University of Medicine and Pharmacy Timisoara, Romania (No. 15/20.03.2020). All subjects included in this study provided informed consent before participation.

To proceed with the validation of the Romanian version of the Toronto Empathy Questionnaire, written permission for use and validation was requested and received from the authors (Prof. Nathan Spreng, McGill University, Montreal, QC, Canada, documents available upon reasonable request). The Toronto Empathy Questionnaire (TEQ) validation was performed in several steps (Figure 1).

### 2.2. Methods, Techniques, and Instruments

The first step consisted of translating the questionnaire validated in English by Spreng et al. [1] to Romanian. The translation was made by an International English Language Testing System (IELTS) certified psychiatrist, with work experience in both clinical and academic settings in the United Kingdom. The questionnaire was subsequently translated back into English by a second psychiatrist, which also benefits from proficient English language skills. The original English version was then compared with the backward-translated version. Minor corrections were made to the Romanian version. The pretest of the questionnaire was performed on ten volunteers working in a panel and represented the second validation step, which aimed to provide a clear and easy understanding of each question. This method was similar to the one employed by other research groups validating assessment scales for the Romanian population [26].

The panel work resulted in some minor improvements in the wording of certain questions. The Romanian version of the Toronto Empathy Questionnaire (TEQ) is presented in Appendix A.

The third step represented the internal and external reliability analysis, conducted on 67 undergraduate medical students from the “Victor Babes” University of Medicine and Pharmacy Timisoara, Romania, who were tested and then re-tested 21 days after the first round. Cronbach’s alpha reliability coefficient was determined for the main parts of the questionnaire. The external consistency (or test-retest) was assessed with the intraclass correlation coefficient.

### 2.3. Research Population and Sample

Next, in step 4, the students of the 4th, 5th, and 6th year General Medicine section of Victor Babes University of Medicine and Pharmacy, Timisoara, Romania, were invited to participate in a survey. The data were collected from March to May 2021. The first section of the questionnaire included the introduction of the survey and a consent form. Besides specific empathy questions, the questionnaire contained demographic questions, such as gender, year of study, and students’ average grades. A Google Play application (android and iOS) or a desktop version on a web platform was developed to help students complete the questionnaire. To ensure total anonymity, a series of alphanumeric codes have been generated so that each student receives a random cipher, which allows him/her to access the test. In addition to this application, a desktop version on a web platform (https://timsonet.ro, accessed on 1 November 2021) was also made available.

### 2.4. Data Processing

A computerized database was created using the Stata program version 16.1 (StataCorp, College Station, TX, USA). The results are presented as absolute and relative frequencies. Continuous variables are presented as mean and standard deviation (SD) or medians and interquartile ranges (IQR). A *p*-value < 0.05 was considered statistically significant. The psychometric properties of the Romanian TEQ were examined in terms of its validity, reliability, and goodness of fit using structural equation modeling. For Cronbach’s alpha level > 0.7 was considered acceptable. The test-retest reliability was assessed using theintraclass correlation coefficient (ICC), with a value ≥ 0.3 considered as acceptable. Principal component analysis (PCA) was conducted using all 16 questions of the TEQ, with the orthogonal rotation (varimax). Kaiser-Meyer-Olkin (KMO) coefficient (>0.7) and the Bartlett Sphericity test (*p* > 0.05) were used to test the suitability of the data for conducting PCA. For confirmatory factor analysis (CFA), the Tucker Lewis index (>0.9), Comparative Fit Index (>0.9), Root Mean Square Error of Approximation (<0.08), and Standardized Root Mean Square Residual (<0.08) were used to assess the model fit.

## 3. Results

Internal and external reliability were calculated overall and for positively and negatively worded items separately, in step 3. Table 1 includes the values of Cronbach’s alpha as a measure of internal reliability and the interclass correlation coefficient as a measure of external reliability. All measures exceeded the 0.7 thresholds, except for the Cronbach’s alpha on negative items, which equals 0.443.

Descriptive statistics for each question for the test-retest time points are presented in Table 2.

In step 4, when the questionnaire was applied to the populational sample of *n* = 649 students, the total score had a median of 49, with an interquartile range of 7. Median (interquartile range) scores for positively and negatively worded items were 24 (5), respectively 25 (5). Female students obtained significantly higher scores than male students, with a medium effect size for total scores and positively worded items and a small to medium effect size for negatively worded items. For other demographic factors taken into consideration (age category, year of study, and last year’s final grade), no significant differences were discovered (Table 3).

The Kaiser-Meyer-Olkin (KMO = 0.790) verified the sampling adequacy for the analysis, and Bartlett’s test of sphericity (*p* < 0.001) indicated the correlations between items were sufficiently large for PCA. Four components had eigenvalues over Kaiser’s criterion of 1 (Figure 2), but only the first three were kept, which explained 34.2% of the variance in combination. The scree plot and the eigenvalues are presented in Figure 2. The factor loadings after varimax rotation and the percentage of explained variance are presented in Table 4, along with the eigenvalues for the rotated component matrix. Using the threshold of 0.3 for factor loadings, the items that cluster on the same components suggest that component 1, which includes questions 3, 5, 6, 13, 16, represents empathic concern/pro-social helping behavior, component 2 which includes questions 2, 4, 10, 12 represents indifference/inappropriate sensitivity, component 3 which includes questions 7, 11, 14, 15 represents disregard/dismissive attitude. Component 1 includes positively worded questions, and components 2 and 3 include negatively worded questions. Questions 1, 8, and 9 are not included in either component.

Figure 3 presents the structural equation modeling (SEM) for the three components that include 13 questions. R square for the equation level for the goodness of fit is 0.91. Confirmatory factor analysis acknowledged that the 3-factor model had satisfactory levels for Comparative fit index (0.749) and Tucker-Lewis index (0.699), Root Mean Square Error of Approximation (0.069), and Standardized Root Mean Square Residual (0.09).

## 4. Discussion

For a future doctor, medicine implies scientific knowledge, but also empathy—an essential emotional experience. The main goal of our profession is to treat the ailing, by empathic, not just medical care. When one feels that the medical professional understands their condition and problems, they will feel more comfortable in any given medical situation. Currently, there is no validated tool to collect empathy data for medical students in Romania, to the best of our knowledge. The present study aimed to validate and adapt the Toronto Empathy Questionnaire, initially developed by Spreng et al., to the Romanian language [1].

Our study introduces TEQ to measure the empathy levels in the Romanian population by examining its psychometric properties through a test-retest method and by using a focus group. The internal reliability of this test reported by other countries, as shown by the value of α, was 0.85 in Canada [1], 0.72 in Greece [27], 0.71 in a French study on general practitioners [28]. Therefore, our data suggest the excellent reliability of TEQ, with a Cronbach’s alpha of 0.727.

Similarly, external reliability, measured by ICC in our study, was 0.776, similar to reports from China (0.78) [8] and the Czech Republic (0.84) [29] for the same instrument.

### 4.1. Components of the Confirmatory Analysis

The first component, represented by prosocial behavior comprises 5 items (3, 5, 6, 13, 16) that express a preoccupation toward the rights, feelings, and welfare of others. All the actions destined to benefit one or more people, other than self-help, recomforting, cooperation—are part of the prosocial behaviors category. Motivations that underlie this behavior can be selfish (improving self-image), have reciprocal benefits (doing something nice for someone else, in order for them to do the same for you) or for altruistic reasons (acting on something out of empathy).

The second component consisted of *inappropriate sensitivity* and grouped 4 items (2, 4, 10, 12), which represent a lack of emotional response. Innately, being happy for someone else or understanding their upset are indispensable abilities for the development and progress of a strong and healthy social rapport. The medical profession requires the presence of this component to establish an adequate doctor-patient therapeutic relationship.

The third component that resulted from our study is represented by a *dismissive attitude and comprises* a further 4 items (7, 11, 14, 15). This component is characterized by a lack of acceptance and of understanding the emotions or problems experienced by others. Rejection and disinterest regarding other people’s difficulties constitute an issue, especially when it comes to doctors, whose main roles are to identify issues and problem-solve. While different levels of empathy were obtained in various studies across the world, it is commonly agreed that healthcare professionals should have the ability to adopt their patients’ perspectives and be compassionate. A study developed in France determined that clinical experience was not proportional to the practitioner’s level of empathy [28]. Recently, in a cross-sectional study among GPs in Denmark [30], empathy did not differ with age or years of practice, but with personal characteristics such as employment outside the clinic and consideration of physician-patient relationship and interaction with colleagues.

### 4.2. Differences Regarding the Undergraduate Year of Study Participants

Although others have found higher levels of empathy in students graduating medical school (6th year) as compared to students entering medical studies (1st year) [31], our study that included senior years of medical school (4th to 6th year) did not find a difference of empathy. One explanation is that sensitization to others was taught in the fundamental disciplines included in the first half of the medical curriculum. Another research developed in Turkey [32] investigated whether the values of TEQ varied amongst medical students as a result of their year of study and specific characteristics. The results showed that TEQ scores increased progressively until the fourth year, although not statistically significant, and then decreased. Also, the same study that included 300 students determined a statistically significant difference in TEQ scores between genders, as women showed higher scores. The same conclusion was started by a team of researchers in Malaysia [33] that studied empathy levels amongst medical students, where the female students scored higher on the TEQ than the male participants, but there were no significant differences determined by the year of study. In a study investigating empathy, a Serbian team of researchers [34] included some 178 first-year medical students and 185 senior medical students. The mean score on the TEQ was 45.23 ± 7.02, and the levels of empathy did not differ between the senior and first-year students.

Pohontschet al. [35] have found through a semi-structured interview that the development and expression of empathy were related to integrating patient contact early on in the curriculum and focusing more on teaching adequate communication and interaction behaviors.

### 4.3. Differences Regarding the Gender of Study Participants

In our study, women have obtained significantly higher scores overall, but also in positive and negative worded questions. As already discussed above, this result was not a constant finding, though most of the reports acknowledge women with higher empathy [29,33,34]. Schulte-Ruther et al. [36] suggested that women and men rely on different strategies when assessing their own emotions in response to other people, with women enhancing reliance on the human mirror neuron system and men on the theory of mind associated areas.

Our findings are consistent with other recent publications that supported a 3-factor structure [8,37], though the structure of the components is different. Other authors haveobtained a one-dimensional structure, by excluding especially questions containing double negation [29,32,37].

### 4.4. Excluded Questions

In current validation, the statistics associated with confirmatory factor analysis show a satisfactory confirmation of the model according to the literature [38,39] and to similar models constructed from TEQ [8]. For the Romanian version of TEQ, questions 1, 8, 9 were not included in either of the components nor in the structural equation model. Questions 1, 8, and 9 contributed to latent components retained in the analysis, with factor loadings smaller than the threshold of 0.3. Each of them contributed to other latent components that were not retained in the final analysis due to Kaiser’s criteria. Although these questions are contributing to the questionnaire, their influence is minimal for the 3 components retained in this validation. The structural equation modeling, which tests the relationship between the retained latent variables and the retained observed variables contributing to them, showed an adequate model fitting with values of the Tucker Lewis index, Comparative Fit Index, Root Mean Square Error of Approximation, and Standardized Root Mean Square Residual in the ranges requested by the developers of the technique [40].

Question 1 was also excluded from other validations of TEQ [29] and the reason behind exclusion might be that question 1 is the only item exploring emotional sharing. In the Chinese validation of TEQ, questions 8 and 9 formed a distinct component named by Xu et al. “negative empathy” [8].

According to Eisenberget al. [41], empathy is the driver for prosocial behavior. In a longitudinal follow-up, Eisenberg et al. [42] have shown that prosocial disposition in adulthood was related to empathy and prosocial behavior at much younger ages.

The motivation behind our study was triggered by the following questions, keeping in mind that a medical career takes years to develop: why does an evaluation of empathy matter? Why does a validated empathy scale for the Romanian population matter? These issues must be addressed by keeping in mind the extensive knowledge base that is built during university studies, subsequent residency training, postgraduate courses, which have been practiced for years, or practical skills achieved through laboratory work, in operating rooms, or by interacting with thousands of patients—these represent essential abilities that are required in the practice of all medical doctors. At the same time, skills such as communication, empathy, persuasion, the ability to adapt and work as part of a team, are just as valuable. The evaluation of empathy, alongside other abilities, should be made at the start of the medical career, for Academics involved in the teaching and training of students to be better able to develop these skills, since we know that the higher placement a person achieves in their career, the more that these personality dimensions tend to matter. Besides, the future professional development of students, faculty members are also involved in supporting their mental health and wellbeing, which is imperative, since there are various studies that have posited that empathy tends to wax and wane throughout years of strenuous medical training [43,44]. In view of this, there should be a larger focus on the counseling services that exist in medical universities, to be better able to offer individualized, tailored psychological interventions, decreasing the stigma of mental health problems in young people, with a focus on the ever-increasing risk of suicide and addictive behaviors among students, or by offering social skills training for specific issues [45].

## 5. Conclusions

Our study reported that the TEQ is a valuable and reliable instrument for an accurate evaluation of the empathy levels of Romanian medical students. The results from this validation study are not dissimilar to other published TEQ translations, by supporting a three-component questionnaire, with the following items: prosocial helping behavior, inappropriate sensitivity, dismissive attitude. This study has several limitations. First, the representativeness of the sample is a limitation since a convenience sample was used. The TEQ was not explicitly designed to assess the empathy of medical university students, representing a young, healthy, and highly educated population. This could influence the evaluation of their empathy, and it may potentially limit the generalizability of the results. Also, we used a self-completed questionnaire, and the answers might be affected by response bias. Another limitation was that we did not evaluate the convergent validity of the instrument. Future studies are needed to test the questionnaire among various populations further to produce more valid results.

We encourage the use of the TEQ in future studies to further test the empathy levels of medical students/medical professionals from other Romanian universities, as a means to reliably measure the need for tailored interventions that would address medical staff burnout and would, inherently, improve patient care and overall quality of health services.

## Figures and Tables

**Figure 1 ijerph-18-12871-f001:**
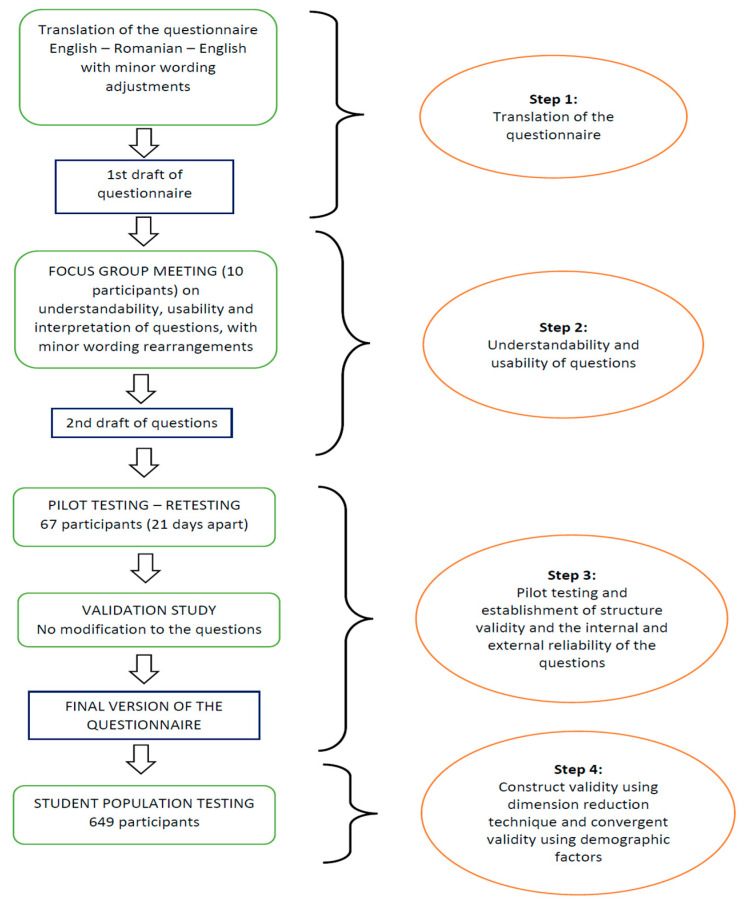
Flow diagram for the steps undertaken for validity testing of Toronto Empathy Questionnaire among Undergraduate Medical Students.

**Figure 2 ijerph-18-12871-f002:**
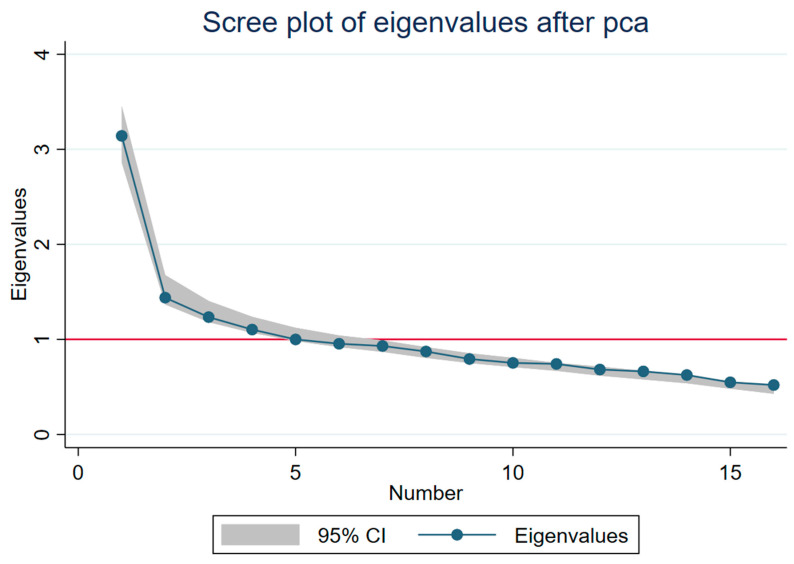
Scree plot of eigenvalues (95% confidence interval) after PCA.

**Figure 3 ijerph-18-12871-f003:**
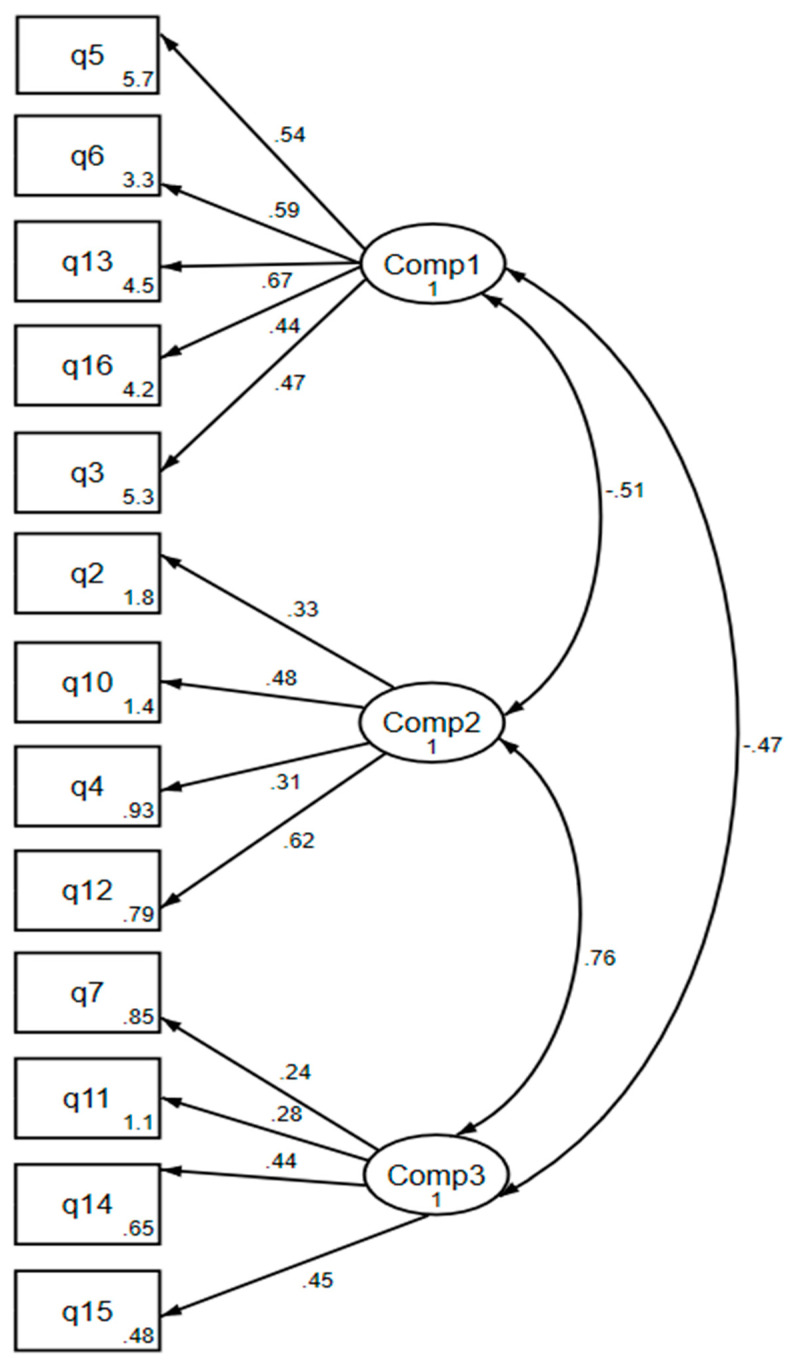
The confirmatory factor analysis of the Romanian version of TEQ with a 3-factor structure. *Legend: numbers on the double-headed arrows represent the covariance between components, numbers on the single-headed arrows represent the standardized factor loadings. the components are: component1 = pro-social helping behavior, indifference, component2 = inappropriate sensitivity, component3 = dismissive attitude.*

**Table 1 ijerph-18-12871-t001:** Internal and external reliability (*n* = 67).

	No. of Items	Cronbach’s Alpha	ICC (Test-Retest)
Overall	16	0.727	0.776
Positively worded items	8	0.745	0.728
Negatively worded items	8	0.443	0.742

**Table 2 ijerph-18-12871-t002:** Descriptive statistics for the TEQ items (test-retest) (*n* = 67).

Item	Mean1 ± SD1	Median1 (Lower-Upper Quartile)	Mean2 ± SD2	Median2 (Lower-Upper Quartile)
1. When someone else is feeling excited, I tend to get excited too.	2.91 ± 0.73	3 (2–3)	3.01 ± 0.75	3 (3–4)
2 *. Other people’s misfortunes do not disturb me a great deal.	1.52 ± 0.99	1 (1–2)	1.39 ± 0.94	1 (1–2)
3. It upsets me to see someone being treated disrespectfully.	3.62 ± 0.62	4 (3–4)	3.57 ± 0.63	4 (3–4)
4 *. I remain unaffected when someone close to me is happy.	0.81 ± 0.82	1 (0–1)	0.76 ± 0.92	1 (0–1)
5. I enjoy making other people feel better.	3.58 ± 0.52	4 (3–4)	3.49 ± 0.56	4 (3–4)
6. I have tender, concerned feelings for people less fortunate than me.	2.79 ± 0.89	3 (2–3)	2.89 ± 0.72	3 (2–3)
7 *. When a friend starts to talk about his\her problems, I try to steer the conversation towards something else.	0.55 ± 0.74	0 (0–1)	0.58 ± 0.70	0 (0–1)
8. I can tell when others are sad even when they do not say anything.	3.25 ± 0.61	3 (3–4)	3.15 ± 0.66	3 (3–4)
9. I find that I am “in tune” with other people’s moods.	2.24 ± 0.76	2 (2–3)	2.27 ± 0.77	2 (2–3)
10 *. I do not feel sympathy for people who cause their own serious illnesses.	1.33 ± 1.05	1 (1–2)	1.18 ± 0.98	1 (0–2)
11 *. I become irritated when someone cries.	0.91 ± 0.95	1 (0–1)	1.04 ± 1.00	1 (0–2)
12 *. I am not really interested in how other people feel.	0.73 ± 0.93	1 (0–1)	0.85 ± 1.06	1 (0–1)
13. I get a strong urge to help when I see someone who is upset.	3.30 ± 0.72	3 (3–4)	3.24 ± 0.74	3 (3–4)
14 *. When I see someone being treated unfairly, I do not feel very much pity for them.	0.85 ± 1.18	1 (0–1)	1.00 ± 1.18	1 (0–1)
15 *. I find it silly for people to cry out of happiness.	0.28 ± 0.71	0 (0–0)	0.39 ± 0.83	0 (0–0)
16. When I see someone being taken advantage of, I feel kind of protective towards him\her	3.15 ± 0.78	3 (3–4)	3.06 ± 0.62	3 (3–3)

Note: * Items are reversed. SD = standard deviation.

**Table 3 ijerph-18-12871-t003:** Total, positive, and negative total scores per demographic factors (*n* = 649).

Demographic Factors		Overall	Positively Worded Items	Negatively Worded Items
TEQ total score		49.0 (45.0–52.0)	24.0 (22.0–27.0)	25.0 (22.0–27.0)
Gender	M	46.0 (42.0–48.0)	23.0 (20.0–24.0)	23.0 (21.0–25.0)
	F	50.0 (46.0–53.0)	25.0 (23.0–27.0)	25.0 (23.0–27.0)
	*p*-value *	*p* < 0.001	*p* < 0.001	*p* < 0.001
Age category	21–22 years	48.0 (45.0- 52.0)	24.0 (22.0- 26.0)	24.0 (22.0–27.0)
	23–24 years	50.0 (46.0–52.5)	25.0 (23.0–27.0)	25.0 (23.0–27.0)
	25+ years	49.0 (46.0–52.0)	24.5 (22.0–27.0)	25.0 (22.0–27.0)
	*p*-value *	0.095	0.183	0.112
Year of study	4	48.0 (45.0–52.0)	24.0 (22.0–27.0)	24.0 (22.0–27.0)
	5	49.0 (46.0–53.0)	25.0 (22.0–27.0)	25.0 (22.0–27.0)
	6	49.0 (46.0–52.0)	24.5 (22.0–26.0)	25.0 (23.0–27.0)
	*p*-value *	0.637	0.478	0.960
Last year’s final grade	7 or below	48.0 (44.0–52.0)	24.0 (22.0- 27.0)	25.0 (23.0–27.0)
	8 or above	49.0 (45.0- 52.0)	24.0 (22.0–27.0)	25.0 (22.0–27.0)
	*p*-value *	0.637	0.478	0.960

* Mann-Whitney test.

**Table 4 ijerph-18-12871-t004:** Factor loading for the three components after varimax rotation.

Variable	Comp1	Comp2	Comp3
1. When someone else is feeling excited, I tend to get excited too.	0.1403	−0.1704	0.0725
2 *. Other people’s misfortunes do not disturb me a great deal.	−0.0148	**0.4837**	−0.2192
3. It upsets me to see someone being treated disrespectfully.	**0.4692**	−0.0438	−0.0722
4 *. I remain unaffected when someone close to me is happy.	0.0911	**0.3010**	0.0719
5. I enjoy making other people feel better.	**0.4367**	0.0432	0.1634
6. I have tender, concerned feelings for people less fortunate than me.	**0.4026**	−0.0928	0.0200
7 *. When a friend starts to talk about his\her problems, I try to steer the conversation towards something else.	0.0563	−0.0170	**0.3607**
8. I can tell when others are sad even when they do not say anything.	0.0724	0.3633	−0.4426
9. I find that I am “in tune” with other people’s moods.	0.0290	0.0687	−0.0681
10 *. I do not feel sympathy for people who cause their own serious illnesses.	0.0031	**0.4947**	0.0839
11 *. I become irritated when someone cries.	0.0215	−0.0313	**0.4777**
12 *. I am not really interested in how other people feel.	−0.0348	**0.4151**	0.2161
13. I get a strong urge to help when I see someone who is upset.	**0.4820**	0.0653	0.0212
14 *. When I see someone being treated unfairly, I do not feel very much pity for them.	−0.0444	0.1952	**0.4116**
15 *. I find it silly for people to cry out of happiness.	−0.0185	0.1896	**0.3406**
16. When I see someone being taken advantage of, I feel kind of protective towards him\her	**0.3915**	0.0248	−0.1034
Eigenvalues	2.3	1.63	1.53
Percentage of variance	14.4	10.2	9.6
alpha	0.668	0.435	0.389

Note: * Items are reversed.

## Data Availability

The datasets used and/or analyzed during the current study are available from the corresponding author on reasonable request.

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
