# Peer review of "Validation of the Romanian Version of the Toronto Empathy Questionnaire (TEQ) among Undergraduate Medical Students"

_ijerph, 2021, doi:10.3390/ijerph182412871_

Round 1

Reviewer 1 Report

Dear Authors,

Please find below my comments and recommendations for improving your paper:

  1. Abstract: Please clearly specify the method of translation and validation. There are more methods in the specialty literature and it is important to mention which one you used (as for example at sub-section 2.1. of https://www.mdpi.com/1648-9144/56/8/409). What you mentioned as ‘several steps’ is too general.
  2. I did not find the Romanian translation in the appendix. It must be there for your audience to use and to increase the visibility of your work.
  3. Provide more reasoning for excluding questions 8&9.
  4. This article has an issue of methodology. If your aim was to validate the Romanian translation (see the title), a representative sample (not a conventional one) should have been used. If your aim was to test Romanian medical students’ empathy (see Conclusions), then you should have used the version of a validated language, like English (which medical practitioners, especially Romanian students, know for sure) Consequently, you definitely need to change the title, the abstract and to express very clearly your aim in the Introduction and Methodology. 

I look forward to the revised version.

Yours faithfully,

Author Response

Dear Reviewer 1,

Thank you for your pertinent observations that we have addressed in the following paragraphs.

  1. Abstract: Please clearly specify the method of translation and validation. There are more methods in the specialty literature and it is important to mention which one you used (as for example at sub-section 2.1. of https://www.mdpi.com/1648-9144/56/8/409). What you mentioned as ‘several steps’ is too general.

Thank you for your recommendation. We have now made the necessary amendments in our manuscript, in the format that you have indicated.

  1. I did not find the Romanian translation in the appendix. It must be there for your audience to use and to increase the visibility of your work

Thank you for highlighting this oversight. We have included this in the appendix A.

  1. Provide more reasoning for excluding questions 8&9.

Thank you for your observation. Please allow us to clarify. Questions 1, 8 and 9 contributed to latent components retained in the analysis, with factor loadings smaller than the threshold of 0.3. Each of them contributed to other latent components that were not retained in the final analysis due to Kaiser's criteria. Although these questions are contributing to the questionnaire, their influence is minimal for the 3 components retained in this validation. The structural equation modeling, which tests the relationship between the retained latent variables and the retained observed variables contributing to them, showed an adequate model fitting with values of the Tucker Lewis index, Comparative Fit Index, Root Mean Square Error of Approximation and Standardized Root Mean Square Residual in the ranges requested by the developers of the technique(Beran, T.N., Violato, C. Structural equation modeling in medical research: a primer. BMC Res Notes 3, 267 (2010). https://doi.org/10.1186/1756-0500-3-267;

StataCorp. 2021. Stata: Release 16. Statistical Software. College Station, TX: StataCorp LLC.).

  1. This article has an issue of methodology. If your aim was to validate the Romanian translation (see the title), a representative sample (not a conventional one) should have been used. If your aim was to test Romanian medical students’ empathy (see Conclusions), then you should have used the version of a validated language, like English (which medical practitioners, especially Romanian students, know for sure) Consequently, you definitely need to change the title, the abstract and to express very clearly your aim in the Introduction and Methodology.

Thank you for your suggestion. As stated in the title and body of the manuscript, our aim was to validate the questionnaire, in order for it to be used in subsequent research in a project aimed at medical students, therefore, the validation was performed specifically in this population.

Your proposal to use the English validated questionnaire for Romanians is not common practice. Testing the level of understanding of English in responders is out of the scope of the validation of a questionnaire in another language. Using only students that understand English at a higher level will introduce a selection bias that cannot be accounted for. Because the performance of a particular questionnaire in a given population may not reflect its performance in a different population due to language particularities, for proper validation, the questionnaire must be translated and used as such.

Supporting the explanations from above, please see the methodology sections included in the references from the current manuscript, which describe having used the same instrument translated into Chinese [8], Greek [26], Czech [28], Korean [36], and the following additional references, where the different questionnaires were validated in selected populations (students), using local language after translation (not the original language of the questionnaire):

  • Manuela Martínez-Lorca, Alberto Martínez-Lorca, Juan José Criado-Álvarez, Mª Dolores Cabañas Armesilla, José M Latorre, The fear of COVID-19 scale: Validation in spanish university students, Psychiatry Research, Volume 293, 2020, 113350, ISSN 0165-1781, https://doi.org/10.1016/j.psychres.2020.113350.
  • Stanisavljevic D, Trajkovic G, Marinkovic J, Bukumiric Z, Cirkovic A, et al. (2014) Assessing Attitudes towards Statistics among Medical Students: Psychometric Properties of the Serbian Version of the Survey of Attitudes Towards Statistics (SATS). PLOS ONE 9(11): e112567.https://doi.org/10.1371/journal.pone.0112567
  • Bourdier L, Lalanne C, Morvan Y, Kern L, Romo L, Berthoz S. Validation and Factor Structure of the French-Language Version of the Emotional Appetite Questionnaire (EMAQ). Front Psychol. 2017;8:442. Published 2017 Mar 23. doi:10.3389/fpsyg.2017.00442
  • Chard CA, Hilzendegen C, Barthels F, Stroebele-Benschop N. Psychometric evaluation of the English version of the Düsseldorf Orthorexie Scale (DOS) and the prevalence of orthorexia nervosa among a US student sample. Eating and Weight Disorders-Studies on Anorexia, Bulimia and Obesity. 2019 Apr;24(2):275-81.

You can also please see the attachment.

Reviewer 2 Report

In your introduction please present the article outline. Conclusions: Give a summary of the findings and present the most salient outcomes in light of the theme and literature.

Author Response

Dear Reviewer 2,

Thank you for your pertinent observations that we have addressed in the following paragraphs.

Comments and Suggestions for Authors

In your introduction please present the article outline. Conclusions: Give a summary of the findings and present the most salient outcomes in light of the theme and literature.

Thank you for your suggestion, the outline of the paper has now been clarified in the introduction. We have now also amended the conclusions, to include a clearer take-home message.

You can also please see the attachment.

Reviewer 3 Report

The publication deals with an important topic, and overall created a good impression. I agree with the limitations listed by the authors. During the examination there were a number of small rather technical issues that need to be finalized.

  1. Keywords:TEQ - decode
  2. align text
  3. there must be a space between the word  and [
  4. insert at the beginning of materials and methods - This study was approved by the Research Ethics Committee of the “Victor Babes”
    University of Medicine and Pharmacy Timisoara, Romania (No. 15/20.03.2020). All subjects included in this study provided informed consent before participation.
  5. Table 4 does not indicate what is meant by Table 4 does not indicate *
  6. Pay attention - you have old literature, update it where you can

Author Response

Dear Reviewer 3,

Thank you for your kind review and pertinent observations that we have addressed in the following paragraphs.

Comments and Suggestions for Authors

The publication deals with an important topic, and overall created a good impression. I agree with the limitations listed by the authors. During the examination there were a number of small rather technical issues that need to be finalized.

1. Keywords:TEQ– decode

Thank you for noticing this oversight on our part. The abbreviation has now been explained.

2. align text

All misalignments have now been corrected.

3. there must be a space between the word and [

A space has now been inserted between the word and [

4. insert at the beginning of materials and methods - This study was approved by the Research Ethics Committee of the “Victor Babes” University of Medicine and Pharmacy Timisoara, Romania (No. 15/20.03.2020). All subjects included in this study provided informed consent before participation.

This paragraph was moved according to your suggestion.

5. Table 4 does not indicate what is meant by *

The symbol * has now been explained.

6. Pay attention - you have old literature, update it where you can

Thank you for this suggestion. We have now included several new references.

You can also please see the attachment.

Round 2

Reviewer 1 Report

Dear Authors,

Congrats on the improvement of the paper. Additionally, I would make further recommendations such as:

  1. divide the 2nd section into sub-sections and clearly present: 2.1. scope, importance, and objectives of the research; 2.2. methods, techniques, and instruments; 2.3. research population and sample; 2.4. data processing
  2. reference the methodology of other research articles validating scales in Romanian, such as https://www.mdpi.com/1648-9144/56/8/409/htm 
  3. divide the 4th section into 4 sub-sections, one for every constitutive part and one for multivariate analysis (e.g. info referring to gender differences, year of study, etc.)
  4. take out 'limitations' from section 4 and put it within conclusions which shall start with your strengths first.

I look forward to hearing your paper has been published.

Yours faithfully,

Author Response

Thank you for your suggestions. We have now made the necessary amendments in our manuscript, in the format that you have indicated. We have also added the suggested reference.